# Covid-19-Associated Coagulopathy: Biomarkers of Thrombin Generation and Fibrinolysis Leading the Outcome

**DOI:** 10.3390/jcm9113487

**Published:** 2020-10-28

**Authors:** Marco Ranucci, Clementina Sitzia, Ekaterina Baryshnikova, Umberto Di Dedda, Rosanna Cardani, Fabio Martelli, Massimiliano Corsi Romanelli

**Affiliations:** 1Department of Cardiovascular Anesthesia and Intensive Care, IRCCS Policlinico San Donato, I-20097 San Donato Milanese, Milan, Italy; ekaterina.baryshnikova@gmail.com (E.B.); umbertodidedda@gmail.com (U.D.D.); 2Department of Biomedical Sciences for Health, Chair of Clinical Pathology, University of Milan, I-20133 Milan, Italy; clementina.sitzia@unimi.it (C.S.); mmcorsi@unimi.it (M.C.R.); 3Operative Unit of Clinical Pathology SMEL-1, Department of Pathology and Laboratory Medicine, IRCCS Policlinico San Donato, I-20097 San Donato Milanese, Milan, Italy; rosanna.cardani@grupposandonato.it; 4Molecular Cardiology Laboratory, IRCCS Policlinico San Donato, I-20097 San Donato Milanese, Milan, Italy; fabio.martelli@grupposandonato.it

**Keywords:** COVID-19, sepsis, pulmonary thromboembolism, thrombin generation, fibrinolysis, heparin

## Abstract

*Background:* Coronavirus Disease 2019 (COVID-19)-associated coagulopathy is characterized by a prothrombotic state not yet comprehensively studied. We investigated the coagulation pattern of patients with COVID-19 acute respiratory distress syndrome (ARDS), comparing patients who survived to those who did not. *Methods:* In this prospective cohort study on 20 COVID-19 ARDS patients, the following biomarkers were measured: thrombin generation (prothrombin fragment 1 + 2 (PF 1 + 2)), fibrinolysis activation (tissue plasminogen activator (tPA)) and inhibition (plasminogen activator inhibitor 2 (PAI-2)), fibrin synthesis (fibrinopeptide A) and fibrinolysis magnitude (plasmin–antiplasmin complex (PAP) and D-dimers). Measurements were done upon intensive care unit (ICU) admission and after 10–14 days. *Results:* There was increased thrombin generation; modest or null release of t-PA; and increased levels of PAI-2, fibrinopeptide A, PAP and D-dimers. At baseline, nonsurvivors had a significantly (*p* = 0.014) higher PAI-2/PAP ratio than survivors (109, interquartile range (IQR) 18.1–216, vs. 8.7, IQR 2.9–12.6). At follow-up, thrombin generation was significantly (*p* = 0.025) reduced in survivors (PF 1 + 2 from 396 pg/mL, IQR 185–585 to 237 pg/mL, IQR 120–393), whereas it increased in nonsurvivors. Fibrinolysis inhibition at follow-up remained stable in survivors and increased in nonsurvivors, leading to a significant (*p* = 0.026) difference in PAI-2 levels (161 pg/mL, IQR 50–334, vs. 1088 pg/mL, IQR 177–1565). *Conclusion:* Severe patterns of COVID-19 ARDS are characterized by a thrombin burst and the consequent coagulation activation. Mechanisms of fibrinolysis regulation appear unbalanced toward fibrinolysis inhibition. This pattern ameliorates in survivors, whereas it worsens in nonsurvivors.

## 1. Introduction

COVID-19-associated coagulopathy (CoAC) is a recognized entity that determines morbidity and mortality, especially in patients with acute respiratory distress syndrome (ARDS). CoAC is characterized by elevated levels of fibrinogen and D-dimers [1,2,3,4,5]. Different reports have shown either thrombocytosis [4,6] or mild thrombocytopenia [7,8] with variable changes in activated partial thromboplastin time (aPTT) and prothrombin time (PT) [4,9]. Clinical series [10,11], autopsy reports [12] and computerized tomography angiography [13] have highlighted the clinical consequences of CoAC, represented by a number of thromboembolic complications, especially at the level of the pulmonary circulation. Conversely, hemorrhagic complications are rare, even if patterns of disseminated intravascular coagulation (DIC) have been reported in patients who died due to COVID-19 infection [3]. Even if the CoAC pattern includes some early-phase, thrombotic-type DIC finding, levels of endogenous anticoagulant proteins, as well as platelet count, may be normal. The clinical impact of CoAC is relevant, with a high incidence of thromboembolic complications found in up to 50% of the patients who had been admitted to the intensive care unit (ICU) for over 2 weeks [14]. In this respect, CoAC appears as a peculiar entity presenting an important challenge and therapeutic dilemmas for the clinicians.

At present, a comprehensive analysis of the complex mechanisms underlying CoAC is lacking, with a gap in knowledge with respect to the balance between the different factors regulating thrombin generation, clot formation and fibrinolysis.

The purpose of the present study was to elucidate the mechanism(s) underlying CoAC in patients with COVID-19 ARDS through the measure of coagulation and fibrinolysis markers and to investigate the relationship between CoAC and the outcome of COVID-19 ARDS patients mechanically ventilated in the intensive care unit (ICU).

## 2. Methods

The present study is part of a wide project (COVID-OMICS) prospectively undertaken at the IRCCS Policlinico San Donato at the beginning of the COVID-19 pandemic. The study was approved by the Local Ethics Committee of San Raffaele Hospital (Code: 75/INT/2020) and registered at clinicaltrials.gov (NCT04441502). All surviving patients gave written informed consent. The coagulation arm was planned on 20 COVID-19 ARDS patients admitted to the ICU and mechanically ventilated.

### 2.1. Patient Population and Treatments

Twenty patients were randomly selected within our population of COVID-19 ARDS patients admitted to the ICU and mechanically ventilated. The first patient was admitted on 27 March 2020, and the last on 21 April 2020. During the course of their stay in the ICU, they received variable treatments according to the changing scenario of international recommendations. These included hydroxychloroquine, tocilizumab and steroids (2 mg/kg methylprednisolone for 5 days followed by 0.5 mg/kg in the next days).

Anticoagulation was established according to a local protocol already published [6], with an aggressive regimen of low-molecular-weight heparin (LMWH): 6000 IU b.i.d. (8000 IU b.i.d. if body mass index > 35). Additionally, antithrombin concentrate was applied to correct antithrombin activity values <70%; antiplatelet agents were used (clopidogrel loading dose 300 mg + 75 mg/day) if platelet count was >400,000 cells/μL.

All the patients were sedated with propofol or midazolam and mechanically ventilated under full muscle relaxant dose at baseline; survivors could still be under full mechanical ventilation support or under weaning from mechanical ventilation at the time of follow-up.

### 2.2. Measurements

We measured the following coagulation and fibrinolysis markers: prothrombin fragment 1 + 2 (PF 1 + 2, ELISA LS-F23736 LifeSpan BioSciences, Seattle, WA, USA), a marker of thrombin generation; plasmin–antiplasmin complex (PAP, ELISA LS-F21825 LifeSpan BioSciences, Seattle, WA, USA), a marker of plasmin generation; tissue plasminogen activator (tPA, Elisa DuoSet DY7449-05 R&D Systems, Biotechne, Minneapolis, MN, USA), a marker of fibrinolysis activation; plasminogen activator inhibitor 2 (PAI-2, Elisa DuoSet DY8550-05 R&D Systems, Biotechne, Minneapolis, MN, USA), a marker of fibrinolysis inhibition; and fibrinopeptide A (ELISA LS-F20727 LifeSpan BioSciences, Seattle, WA, USA), a marker of fibrin generation.

Besides these markers, we measured the ratio between PAI-2 (pg/mL) and PAP (ng/mL) as a marker of the balance between fibrinolysis inhibition and fibrinolysis amount.

Additionally, the standard hemostasis and coagulation characterization included the measure of the aPTT, INR, platelet count, fibrinogen, D-dimers and antithrombin (AT) activity. INR and aPTT were assessed using the STA-NeoPTimal 10 and the STA-Cephascreen 10 (Diagnostica Stago, Asnières sur Seine, France), respectively; fibrinogen was measured using the Clauss-based STA-LiquidFib (Diagnostica Stago, Asnières sur Seine, France).

The blood samples were collected at different points in time: at the admission to the ICU and at variable intervals (5–7 days) from the admission to the ICU. For the purposes of the present analysis, we considered two points in time: baseline (admission to the ICU) and follow-up. Follow-up corresponded to the last measure before death for nonsurvivors and to a similar point in time for survivors.

### 2.3. Outcome Assessment

The primary outcome was defined as survival at 45 days from ICU admission or mortality within 45 days from ICU admission. Thromboembolic events at the level of pulmonary circulation were defined on the basis of computerized tomography (CT) when available. No routine assessment of venous thrombosis at other sites was performed.

### 2.4. Statistics

Data are presented as number (%) or median (interquartile range (IQR)). Differences between groups (survivors vs. nonsurvivors) were tested with nonparametric tests (Mann–Whitney U-test), and differences between baseline and follow-up within groups with a Wilcoxon signed rank test. Predictive ability of the different parameters was tested with a receiver operating characteristic (ROC) analysis producing c-statistics and sensitivity, specificity and positive and negative predictive values (PPV and NPV, respectively) for the identified cut-off values. For all the tests, a two-tailed *p* value <0.05 was considered significant. The statistical analyses were conducted using computerized packages (SPPS 13.0, IBM, Chicago, IL, USA; MedCalc, Ostend, Belgium).

## 3. Results

Overall, we reported 8 (40%) survivors and 12 (60%) nonsurvivors. One patient (5%) had a previous cerebrovascular accident associated with atrial fibrillation and was under therapy with direct oral anticoagulants. No cases of known thrombophilia were reported.

Table 1 reports the general characteristics and standard coagulation tests in the patient population. Factors significantly associated with mortality were age and the CT evidence of pulmonary thromboembolism, found in seven (35%) patients all belonging to the nonsurvivor group, with a significant (*p* = 0.015) between-group difference.

Table 2 reports the coagulation parameters related to thrombin generation and fibrinolytic profile of the patient population, at baseline and follow-up. The median time between baseline and follow-up was 17 days (IQR 14–24 days) in survivors and 13 days (IQR 6–22 days) in nonsurvivors (*p* = 0.134). 

At baseline assessment, the only significant (*p* = 0.014) difference between survivors and nonsurvivors was a 10 times higher value of the PAI-2/PAP ratio in nonsurvivors. In both groups, the values for fibrinogen, D-dimers, PAI-2, PF 1 + 2, PAP and fibrinopeptide A were above the normal ranges reported in the literature. Conversely, the value for tPA was within the normal range. At follow-up, nonsurvivors had a significantly higher levels of D-dimers and PAI-2 (*p* = 0.003 and *p* = 0.026, respectively).

There were within-group differences between values at baseline and follow-up. Survivors showed a significant (*p* = 0.025) decrease of PF 1 + 2 and a nonsignificant decrease of fibrinogen, tPA, PAI-2, PAP and PAI-2/PAP ratio. Nonsurvivors showed a nonsignificant increase in D-dimers, PAI-2 and PF 1 + 2, while PAI-2/PAP ratio remained stable.

In the overall patient population, PAP concentrations were strongly and directly correlated with fibrinopeptide A concentrations at baseline (R^2^: 0.98, *p* = 0.001) and moderately correlated at follow-up (R^2^:0.51, *p* = 0.001). Conversely, D-dimers were not dependent on fibrinopeptide A levels (Figure 1).

The association between coagulation markers and other clinical variables was investigated using a Pearson’s correlation matrix. There was no association between age, gender, eGFR and any of the coagulation markers, both at the admission to the ICU and at follow-up.

The ability of the different markers measured at baseline to predict mortality was investigated with an ROC analysis. The only parameters with a c-statistic ≥0.80 were D-dimers and the PAI-2/PAP ratio, with values of 0.813 (95% confidence interval 0.47–0.89, *p* = 0.089) and 0.875 (95% confidence interval 0.60–0.96, *p* = 0.001), respectively. Figure 2 shows the dot plot of distribution for PAI2/PAP ratio and D-dimers. The best cut-off values (best fit between specificity and sensitivity) were found at a level of 1.13 μg/mL for D-dimers and 12.9 for the PAI-2/PAP ratio. These values correspond to sensitivity of 83.3% for D-dimers and 83.3% for PAI-2/PAP ratio and specificity of 62.5% for D-dimers and 87.5% for PAI-2/PAP ratio. Considering a prevalence of mortality of 55% (as recorded for the whole patient population admitted to the ICU in our institution), the PPV and NPV for D-dimers were 73.1% and 75.4%, respectively, while for PAI-2/PAP ratio they were 89.1% and 81.1%, respectively.

## 4. Discussion

Our results provide an interpretation of the already known procoagulant pattern of patients with ARDS due to COVID-19 infection, and, to our knowledge, this is the first investigation of the coagulation profile based on markers of thrombin generation and fibrinolysis. Previous studies with viscoelastic tests had already stressed that the main finding in these patients is an abnormally increased clot firmness, with no signs of hyperfibrinolysis or even fibrinolysis shutdown [6,15,16,17]. However, analyses based on standard or viscoelastic tests remain inconclusive with respect to the nature of this pattern. From this respect, our study suggests an interpretative view of the major factors determining CoAC and their differences in survivors and nonsurvivors.

### 4.1. Thrombin Generation

Thrombin generation cannot be assessed with standard or viscoelastic tests. In the first case, variable values of aPTT and PT have been reported [1,2,3,4,5,6], but their changes obviously reflect even the effects of the antithrombotic therapies. In the second, the reaction times (measured with heparinase) did not show a decreased value suggestive for an increased thrombin generation [15,16]. We addressed thrombin generation by measuring PF 1 + 2, a marker of prothrombin cleavage to thrombin. We found values ranging from 20 to over 2300 pg/mL at baseline (median 442 pg/mL) and from 20 to over 3300 pg/mL at follow-up (median 371 pg/mL). The normal value of PF 1 + 2 in healthy subjects is between 11 and 22 pg/mL; therefore, we found PF 1 + 2 values about 20 times higher than normal. In other models of severe sepsis, median values of 100–200 pg/mL were reported [18]; in our series, thrombin generation was almost double these values. In a model of dengue fever, peak values of PF 1 + 2 were 3–5 times higher than the normal range [19], and similar increases were noticed in hantavirus infection [20]. Therefore, it can be concluded that thrombin generation in coronavirus infection is increased as in other viral infections (including HCV) [21], but to a higher degree. Of notice, thrombin generation behaved differently in survivors and nonsurvivors. At baseline, there were no significant differences between groups; however, in survivors, thrombin generation significantly decreased at follow-up, whereas it remained stable or increased in nonsurvivors.

Thrombin generation during viral infections is usually attributed to a tissue factor (TF)-mediated activation of the extrinsic pathway, with TF release by activated monocytes and endothelial cells. However, a role of the intrinsic pathway cannot be excluded. Factor XI activation was demonstrated in dengue fever [19]. Once thrombin is formed, it can activate factor XI, thus amplifying thrombin generation through a positive feedback. In our study, we did not directly measure activated factor XI or TF, and we cannot separate the role of the two pathways in determining the thrombin burst.

### 4.2. Fibrinogen and Fibrin Generation

Elevated fibrinogen levels were confirmed in our patient population, as already highlighted in other studies [6,15,16]. Fibrinogen levels were decreased by 35% at follow-up in survivors and by only 16% in nonsurvivors. Fibrin generation was addressed by measuring fibrinopeptide A, a marker of fibrinogen cleavage to fibrin. Normal levels of fibrinopeptide A range between 0.1 and 2 ng/mL, with a mean at 0.50 ng/mL [22]. In our series, fibrinopeptide A largely exceeded the upper limit of the normal range, both in survivors and nonsurvivors, at baseline and follow-up, with a trend toward higher values in survivors. Hence, as a logical consequence of the increased thrombin generation, fibrin generation was increased as well, and continued unabated from baseline to follow-up. Fibrinopeptide A increases in patients with bacterial and virus sepsis, as a consequence of the cross-link between inflammation and coagulation. The values found in our series are in the range of those previously found in patterns of bacterial sepsis, severe sepsis and septic shock [23].

### 4.3. Fibrinolysis Activation

Tissue plasminogen activator is a fibrinolytic agent released mainly by endothelial cells as a response to fibrin generation. Its normal values are around 10,000 pg/mL [24,25], but in conditions of systemic inflammatory reaction syndrome or severe sepsis its values are usually higher (>10,000 pg/mL) [26], with reported values up to 50,000–70,000 pg/mL in nonsurvivors [27].

Quite surprisingly, in our series, the median value of tPA was at the lower limits of the normal range both at baseline and follow-up, and both in survivors and nonsurvivors, with only one case reaching 20,000 pg/mL. Therefore, it seems that fibrinolysis was apparently not activated in these patients, despite an increased thrombin (and fibrin) generation.

### 4.4. Fibrinolysis Inhibition

Plasminogen activator inhibitor 2 is a powerful inhibitor of fibrinolysis, released by monocytes. It is usually undetectable in plasma from normal subjects, and it is considered an inhibitor of urokinase-type plasminogen activator acting mainly at an extravascular level [27].

Due to its ability to act at the level of interstitial tissues (including lung interstitium) and its undetectability in normal subjects (except in pregnant women), we measured PAI-2 as a marker of fibrinolysis inhibition. Previous studies highlighted that PAI-2 becomes detectable in septic patients, with values of 500–1000 pg/mL in survivors and up to 30,000 pg/mL in nonsurvivors [27].

In our series, elevated values of PAI-2 were observed, especially in nonsurvivors; at follow-up, nonsurvivors showed a level of PAI-2 6-fold that of survivors, with a significant between-group difference. Therefore, fibrinolysis was inhibited, and the extent of inhibition at follow-up was associated with mortality.

### 4.5. The Net Effect on Fibrinolysis

The markers of fibrinolysis in our series were PAP and D-dimers. PAP is a marker of plasmin formation and of plasmin ability to counteract fibrin generation. Not by chance, we could observe a strong relationship between PAP and the marker of fibrin generation fibrinopeptide A. Levels of PAP are usually greatly increased under conditions of inflammation and sepsis, with levels exceeding 1000 ng/mL [28]. We could only observe a modest increase of PAP with respect to the reported normal range of 19–27 ng/mL [28], more pronounced in survivors than in nonsurvivors. Therefore, fibrinolysis appeared to be in a shutdown condition, as the result of the balance between increased antifibrinolytic agents (PAI-2) and stable fibrinolytic agents (tPA). This shutdown appeared to be more pronounced in nonsurvivors, where the PAI-2/PAP ratio was significantly higher than that in survivors.

Within this scenario, a particular aspect is represented by D-dimer behavior. D-dimers are a fibrin degradation product, and therefore their increase, found both in survivors and (to a larger extent) in nonsurvivors, should be interpreted as marker of increased fibrinolysis. However, contrary to PAP, D-dimers have no correlation with fibrin generation (as represented by fibrinopeptide A). Therefore, their increase cannot be ascribed solely to the increased levels of fibrin. Actually, the source of D-dimer increase in COVID-19 patients is still a matter of debate, and the role of extravascular fibrin degradation (namely in the interstitial and alveolar lung space) has been hypothesized [29,30].

The overall picture that can be drawn from our results is summarized in Figure 3.

In both survivors (Figure 3A) and nonsurvivors (Figure 3B) there is an initial phase characterized by the release of proinflammatory cytokines and consequent burst of thrombin generation (as previously discussed) reasonably mediated by TF release and amplification of the process by intrinsic pathway activation. The endothelial cells show a very modest release of tPA. At this stage, both thrombin generation and tPA release do not differ between survivors and nonsurvivors. Conversely, in nonsurvivors, the release of PAI-2 is higher than in survivors, switching the balance between fibrinolysis stimulation and inhibition toward inhibition. In both groups there are similar and very high levels of substrate (fibrinogen) and an important increase of fibrin generation. However, fibrinolysis appears more efficient (higher PAP values) in survivors than in nonsurvivors. In both cases it is likely that an initial thrombi formation may intervene (elevated D-dimers).

The two pathways clearly diverge at follow-up. In survivors, thrombin generation, PAI-2 and tPA release decrease, as do fibrinogen levels. Fibrinolysis appears maintained and D-dimers are stable. Conversely, in nonsurvivors, thrombin generation and PAI-2 increase and tPA decreases, with a further shutdown of fibrinolysis and an important increase in D-dimers. This is likely to represent a condition where thrombi formation may become uncontrolled and clinically relevant.

### 4.6. Thromboembolism, Mortality and Its Predictors and Therapeutic Implications

The prothrombotic pattern of CoAC has a relevant clinical impact. Other studies already highlighted the high risk of thromboembolic events in these patients [10,12,13,14]. Within this pattern, the definition of the pathway of CoAC from the onset to the final outcome is important providing that (i) an early recognition of patients at high risk of mortality is feasible and (ii) adequate diagnostic measures and therapies may be established.

With respect to the first issue, our data suggest a high predictive ability of the PAI-2/PAP ratio (cut-off at 12.9) shortly after the patient is tracheally intubated and placed under mechanical ventilation and a moderate predictive ability of D-dimers (cut-off at 1.13 μg/mL). Patients with values above these thresholds deserve pulmonary CT scan angiography for early detection of micro/macrothrombi.

The pathophysiological process leading to negative outcome may be divided into three phases. Therapeutic choices are not the purpose of our study, but some possible interventions may be hypothesized based on our time-related data.

The first is the containment of the inflammatory reaction with blunting of the proinflammatory cytokines. The drugs of choice are steroids: a very recent report demonstrated that, in patients under mechanical ventilation, dexamethasone therapy reduces mortality by 35% [31]. The role of steroids is linked to the containment of proinflammatory cytokine release and the consequent monocyte and endothelial activation. The second is the containment of thrombin generation: heparin is the most commonly used drug for this purpose. Anticoagulation in CoAC is a complex issue, but the role of LMWH is certainly of paramount importance. LMWH inhibits thrombin generation by acting on factor Xa, modulates inflammation by binding inflammatory cytokines and neutralizing complement factor C5a, protects the endothelial layer by antagonizing histone-mediated endothelial damage and possibly exerts direct antiviral properties [32]. A recent recommendation of the International Society of Thrombosis and Hemostasis has addressed the use of LMWH and unfractionated heparin (UFH) in COVID-19 patients [33]. Based on the severity of the coagulopathy, and namely on the evidence or risk of thromboembolic events, the LMWH may be increased from prophylaxis to anticoagulation regimen; in patients with severe renal impairment, UFH may be preferred to LMWH. However, UFH has its own limitations: monitoring (aPTT) can possibly be biased by a number of confounders, and there is the possibility of heparin resistance and heparin-induced thrombocytopenia [33]. In this setting, even direct thrombin inhibitors (argatroban and bivalirudin) may be considered. Finally, the third phase is tackling the fibrinolysis shutdown. In presence of elevated levels of fibrinolysis inhibitors, and with documented CT scan pulmonary thromboembolism, it seems reasonable to consider the administration of rtPA to prevent further thromboembolic events and to trigger thrombi dissolution.

Even if we could find associations between coagulation markers and outcome, coagulation imbalance is of course only one of the factors determining the outcome within the complex scenario of CoAC. As already reported in other series, age at the admission to the ICU is the main determinant of mortality. We could not find any association between age and the coagulation markers at admission or follow-up. This suggests that thrombin generation and impaired fibrinolysis exert their negative impact independently of age. However, it is possible that elderly patient may be more sensitive to the combined effects of thrombin burst and fibrinolysis shutdown. This may be due to a higher predisposition to endothelial dysfunction, as well as to an exacerbated innate host response [32].

In conclusion, our results stress the leading role of thrombin generation and, most importantly, fibrinolysis shutdown in determining the environment for pulmonary micro/macrovascular thrombosis and negative outcomes. In terms of therapeutic implications, adequate randomized or case-control trials are necessary to obtain the required evidence of success.

## Figures and Tables

**Figure 1 jcm-09-03487-f001:**
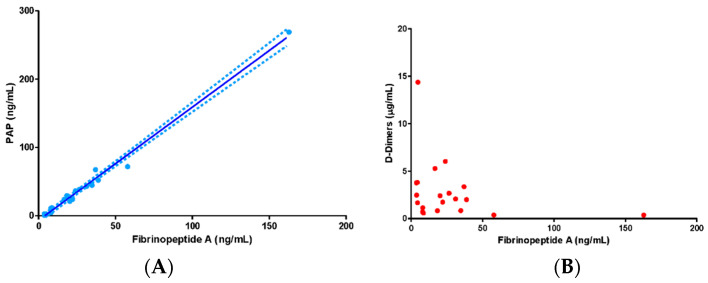
Association between plasma levels of fibrinopeptide A and plasmin–antiplasmin (PAP) complexes (**panel A**) and D-dimers (**panel B**). Data in the text.

**Figure 2 jcm-09-03487-f002:**
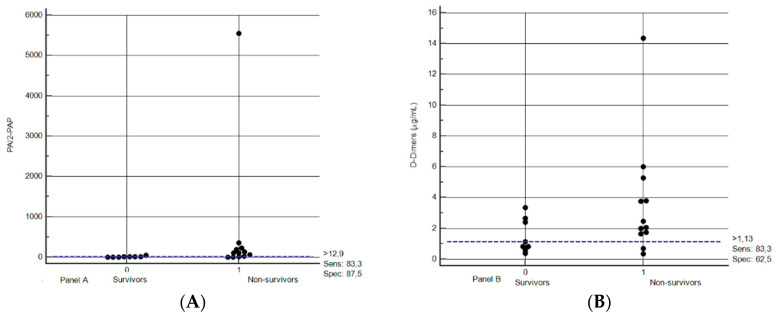
Dot plot distribution of PAI-2/PAP ratio (**panel A**) and D-dimers (**panel B**) in survivors and nonsurvivors. PAI2: plasminogen activator inhibitor 2; PAP: plasmin–antiplasmin complex. Data in the text.

**Figure 3 jcm-09-03487-f003:**
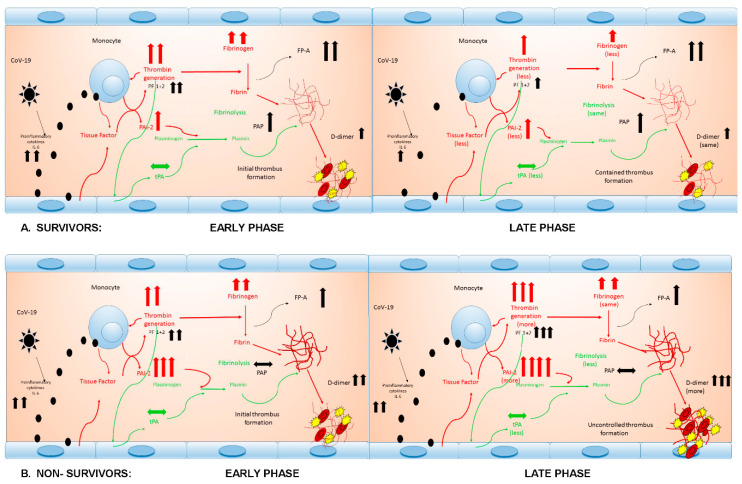
The chain of inflammatory and hemostatic reactions in survivors (**A**) and nonsurvivors (**B**), from early to late phase. FP-A: fibrinopeptide A; IL-6: interleukin-6; PAI-2: plasminogen activator inhibitor 2; PAP: plasmin–antiplasmin complex; PF 1 + 2: prothrombin fragment 1 + 2; t-PA: tissue plasminogen activator. Red lines are prothrombotic; green lines are antithrombotic.

**Table 1 jcm-09-03487-t001:** General profile of the patient population at the admission to the intensive care unit and outcome.

Variable	Overall (*n* = 20)	Survivors (*n* = 8)	Nonsurvivors (*n* = 12)	*p*
Age (years)	63.9 (56.3–71)	59.6 (53.8–63.4	69.4 (63.6–72.3)	0.037
Gender male	16 (80)	7 (87.5)	9 (75)	0.619
Weight (kg)	90.5 (71.2–90)	87.5 (81.2–102.5)	77.5 (66.2–84)	0.075
Body mass index (kg/m^2^)	26.5 (24.6–31.2)	27.7 (26.3–33.5)	25.4 (23.4-30)	0.077
Hypertension	6 (30)	2 (25)	4 (33.3)	0.690
Diabetes	5 (25)	3 (37.5)	2 (16.7)	0.296
Smoking history	3 (15)	2 (25)	1 (8.3)	0.537
Heart disease	3 (15)	0 (0)	3 (25)	0.242
Chronic lung disease	2 (10)	1 (12.5)	1 (8.3)	0.796
e-GFR (mL/min/1.73 m^2^)	101 (58–148)	101 (45–137)	105 (58–159)	0.624
Interleukin-6 (pg/mL)	181 (40–338)	126 (4–126)	182 (43–467)	0.398
aPTT (s)	35.6 (30.3–45.5)	33.5 (28.8–59.8)	38.3 (30.3-45.5)	0.680
Prothrombin activity (%)	86 (78–92)	87 (79–97)	83 (64–91)	0.432
Antithrombin activity (%)	91 (80–98)	91 (81–100)	91 (68-95)	0.276
Platelets (×1000 cells/μL)	254 (148–295)	237 (156–356)	254 (128–295)	0.939
ICU length of stay (days)	13.5 (8–32)	25 (3.5–35)	11.5 (8–17.2)	0.699
Pulmonary TE	7 (35)	0 (0)	7 (58.3)	0.015
(documented at CT scan)				

Data are median (interquartile range) or number (%). aPTT: activated partial thromboplastin time; CT: computerized tomography; eGFR: estimated glomerular filtration rate; ICU: intensive care unit; TE: thromboembolism.

**Table 2 jcm-09-03487-t002:** Thrombin generation and fibrinolytic profile.

Variable	Time	Overall (*n* = 20)	Survivors (*n* = 8)	Nonsurvivors (*n* = 12)	*p*
Fibrinogen (mg/dL)	Baseline	592 (378–745)	485 (288–720)	664 (466–753)	0.215
	Follow-up	476 (297–706)	319 (255–475)	576 (362–737)	0.086
D-dimers (μg/mL)	Baseline	2.02 (0.82–3.6)	0.98 (0.64–2.6)	2.26(1.67–4.90)	0.123
	Follow-up	2.74 (1.55–3.6)	1.61 (1.09–2.4)	3.50(2.66–6.45)	0.003
tPA (pg/mL)	Baseline	4660 (3620–7663)	6323 (4160–7967)	4438 (3482–6387)	0.335
	Follow-up	3622 (1964–10,012)	2233 (1865–7506)	3822 (2751–12,243)	0.123
PAI-2 (pg/mL)	Baseline	400 (50–842)	191 (50–514)	567 (50–2132)	0.184
	Follow-up	423 (50–1238)	161 (50–334)	1088(177–1565)	0.026
PF 1+2 (pg/mL)	Baseline	442 (302–649)	396 (185–585)	442 (302–973)	0.671
	Follow-up	371 (119–763)	237 (120–393) *	557 (106–767)	0.247
PAP (ng/mL)	Baseline	23.9 (3.02–44.2)	33.9 (13.9–61.8)	17.4 (0.97–41.2)	0.164
	Follow-up	23.4 (6.3–55)	27.1 (13.3–95.7)	14.8 (4.2–52)	0.247
FPA (ng/mL)	Baseline	19.3 (5.5–33.8)	23.4 (11–36.4)	12.3 (4.4–29.3)	0.165
	Follow-up	14.1 (5.8–60.2)	20.5 (11.3–63.6)	10.7 (4.4–56.4)	0.247
PAI-2/PAP ratio	Baseline	19.4 (5–125)	8.7 (2.9–12.6)	109 (18.1–216)	0.014
	Follow-up	17.4 (1.8–149)	3.5 (1.8–18.7)	74 (4.2–255)	0.064

FPA: fibrinopeptide A; PAI: plasminogen activator inhibitor; PAP: plasmin–antiplasmin complex; PF: prothrombin fragment; tPA: tissue plasminogen activator. Within-group difference between follow-up and baseline: * *p* = 0.025. Data are given as median (interquartile range).

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
