# Peer review of "Covid-19-Associated Coagulopathy: Biomarkers of Thrombin Generation and Fibrinolysis Leading the Outcome"

_jcm, 2020, doi:10.3390/jcm9113487_

Round 1
Reviewer 1 Report
Authors compared pattern of coagulation in COVID-19 acute respiratory distress syndrome (ARDS) in survivors and non-survivors. The study is well-written, the idea is original, and the topic is of actual interest. However, I have some comments to be addressed:
1) Are there a history of VTE or known thrombophilia in any patients?
2) Does any patient suffer from kidney disease? Please provide further information and if such disease could have affected findings.
3) What about drugs used in patients? Is there any difference between groups? Please discuss the potential impact of anticoagulant and immunomodulatory therapies in coagulation cascade A.
4) Non-survivors are significantly older than survivors. Please discuss how older age could have influenced the results (authors may consult “Coagulopathy, thromboembolic complications, and the use of heparin in COVID-19 pneumonia”, Costanzo L et al. J Vasc Surg Venous Lymphat Disord. 2020; 8:711-716)
5) Please clarify the sentence in line 287-289
6) Do authors believe that unfractionated heparin could have a role in advanced states when powerful intravascular generation of thrombin occurs (line 314-316)?
7) Could the authors hypothesize the mechanism of increased generation of thrombin in COVID respect to other viral infections?
Author Response
Authors compared pattern of coagulation in COVID-19 acute respiratory distress syndrome (ARDS) in survivors and non-survivors. The study is well-written, the idea is original, and the topic is of actual interest.
REPLY:We thank the Reviewer for this comment.
However, I have some comments to be addressed:
1) Are there a history of VTE or known thrombophilia in any patients?
REPLY: We have now looked back at the original files and we have added this to the manuscript (table 1). However, it should be considered that the anamnestic reports are limited by the fact that all the patients entered our ICU in Emergency conditions and already tracheally intubated, the great majority transferred from spoke hospitals.
2) Does any patient suffer from kidney disease? Please provide further information and if such disease could have affected findings.
REPLY: we have now checked renal function at the time of the analyses and we have tested associations between CrCl and coagulation markers. We could not find any association between eGFR and coagulation markers.
3) What about drugs used in patients? Is there any difference between groups? Please discuss the potential impact of anticoagulant and immunomodulatory therapies in coagulation cascade A.
REPLY: all the patients received the same treatment in ICU as reported in the methods. Only one patient received anticoagulant (NOAC) therapy before ICU admission. We could not find any link between pre-admission therapy and mortality. We have now implemented the discussion as suggested for the role of anticoagulants and steroids in coagulation cascade (figure 3, Panel A).
4) Non-survivors are significantly older than survivors. Please discuss how older age could have influenced the results (authors may consult “Coagulopathy, thromboembolic complications, and the use of heparin in COVID-19 pneumonia”, Costanzo L et al. J Vasc Surg Venous Lymphat Disord. 2020; 8:711-716)
REPLY: we have included the Reference within the list. Additionally, we have quoted the recently released ISTH guidelines on anticoagulation in COVID-19. We have now discussed the impact of age on the outcome, both as independent risk factor and as a predisposing factor for CoAC. Of notice, no association between age and coagulation markers was found.
5) Please clarify the sentence in line 287-289
REPLY: done, we thank the Reviewer for spotting an error.
6) Do authors believe that unfractionated heparin could have a role in advanced states when powerful intravascular generation of thrombin occurs (line 314-316)?
REPLY: we have now discussed this at the light of the recently released ISTH guidelines: Thachil J, Juffermans NP, Ranucci M, Connors JM, Warkentin TE, Ortel TL, Levi M, Iba T, Levy JH. ISTH DIC subcommittee communication on anticoagulation in COVID-19. J Thromb Haemost. 2020 Sep;18(9):2138-2144
7) Could the authors hypothesize the mechanism of increased generation of thrombin in COVID respect to other viral infections?
REPLY: actually, many severe viral infections (including dengue fever, hantavirus infection, HCV...) are characterized by an increased thrombin generation. However, we could observe levels of PF 1+2 almost 20 times higher than the normal value, whereas in other infections the increase is usually in the range of 3-5 times higher. In a model of dengue fever, peak values of PF 1+2 were 3-5 times higher than the normal range, and similar increases were noticed in Hantavirus infection. Therefore, it can be concluded that thrombin generation in Coronavirus infection is increased as in other viral infections, (including HCV) but to a higher degree.
About the mechanism of thrombin generation during viral infections, it is usually attributed to a Tissue Factor (TF)-mediated activation of the extrinsic pathway, with TF release by activated monocytes and endothelial cells. However, it cannot be excluded a role of the intrinsic pathway. Factor XI activation was demonstrated in Dengue Fever.Once thrombin is formed, it can activate factor XI, thus amplificating thrombin generation through a positive feedback. In our study we did not directly measure activated factor XI nor TF, and we cannot separate the role of the two pathways in determining the thrombin burst.
We have now widely addressed these concepts in the Discussion.
Reviewer 2 Report
Ranucci and colleagues presented results of biomarkers for in vivo thrombin generation and fibrinolysis in Covid-19 patients (surviving and non-surviving).
As this study only included 20 patients (8 survivors and 12 non-survivors), could the authors present at least the data from figure 2 in dot plot in order for the reader to evaluate the results distribution.
The figure 3 is not helpful and too general. The authors state that increased in vivo thrombin generation results from TF whereas the immunothrombotic state in CoV-19 triggers NETs formation that can activate the intrinsic pathway. More generally, as intrinsic pathway is more implicated in thrombosis than hemostasis it is not possible here to discriminate the TF-dependent activation from the FXII-dependent activation.
Author Response
Ranucci and colleagues presented results of biomarkers for in vivo thrombin generation and fibrinolysis in Covid-19 patients (surviving and non-surviving).
As this study only included 20 patients (8 survivors and 12 non-survivors), could the authors present at least the data from figure 2 in dot plot in order for the reader to evaluate the results distribution.
REPLY: We have now replaced figure 2 (classical ROC analysis) with a dot plot figure
The figure 3 is not helpful and too general. The authors state that increased in vivo thrombin generation results from TF whereas the immunothrombotic state in CoV-19 triggers NETs formation that can activate the intrinsic pathway. More generally, as intrinsic pathway is more implicated in thrombosis than hemostasis it is not possible here to discriminate the TF-dependent activation from the FXII-dependent activation.
REPLY: we thank the Reviewer for this comment. Of course figure 3 cannot summarize all the hypotheses underlying the complex mechanisms of coagulation activation in COVID-19 ARDS patients. It simply a reader-friendly summary of our findings. I totally agree that we cannot discriminate the intrinsic-extrinsic pathway role in determining the thrombin burst. Therefore, we have now included in the Discussion a brief comment on this. Basically, we stressed that thrombin generation during viral infections is usually attributed to a Tissue Factor (TF)-mediated activation of the extrinsic pathway, with TF release by activated monocytes and endothelial cells. However, it cannot be excluded a role of the intrinsic pathway. Factor XI activation was demonstrated in Dengue Fever (van Gorp ECM, Minnema MC, Suharti C, et al. Activation of coagulation factor XI, without detectable contact activation in dengue haemorrhagic fever. Br J Haematol 2001; 113: 94-9). Once thrombin is formed, it can activate factor XI, thus amplificating thrombin generation through a positive feedback. In our study we did not directly measure activated factor XI nor TF, and we cannot separate the role of the two pathways in determining the thrombin burst.
Round 2
Reviewer 1 Report
I would like to congratulate the authors on their work and the thorough review of the manuscript. I have no further comments.